# Indication of Strongly Correlated Electron Transport and Mott Insulator in Disordered Multilayer Ferritin Structures (DMFS)

**DOI:** 10.3390/ma14164527

**Published:** 2021-08-12

**Authors:** Christopher Rourk, Yunbo Huang, Minjing Chen, Cai Shen

**Affiliations:** 1Independent Researcher, Dallas, TX 75205, USA; 2Ningbo Institute of Materials Technology & Engineering Chinese Academy of Sciences, 1219 Zhongguan Road, Zhenhai District, Ningbo 315201, China; 15605219893@163.com (Y.H.); chenminjing@nimte.ac.cn (M.C.); 3College of Materials Science and Opto-Electronic Technology, University of Chinese Academy of Sciences, Beijing 100049, China

**Keywords:** ferritin, quantum dots, layer-by-layer deposition, conductive atomic force microscopy, strong correlations

## Abstract

Electron tunneling in ferritin and between ferritin cores (a transition metal (iron) oxide storage protein) in disordered arrays has been extensively documented, but the electrical behavior of those structures in circuits with more than two electrodes has not been studied. Tests of devices using a layer-by-layer deposition process for forming multilayer arrays of ferritin that have been previously reported indicate that strongly correlated electron transport is occurring, consistent with models of electron transport in quantum dots. Strongly correlated electrons (electrons that engage in strong electron-electron interactions) have been observed in transition metal oxides and quantum dots and can create unusual material behavior that is difficult to model, such as switching between a low resistance metal state and a high resistance Mott insulator state. This paper reports the results of the effect of various degrees of structural homogeneity on the electrical characteristics of these ferritin arrays. These results demonstrate for the first time that these structures can provide a switching function associated with the circuit that they are contained within, consistent with the observed behavior of strongly correlated electrons and Mott insulators.

## 1. Introduction

A quantum dot (QD) is a conducting island of a size comparable to the Fermi wavelength in all spatial directions. The current/voltage (I/V) characteristics of QDs have been previously studied using conductive atomic force microscopy (c-AFM), and it has been shown that QDs can exhibit nonlinear I/V behavior when tested. Unlike conventional linear I/V behavior of bulk materials that is a function of conductivity alone, the nonlinear I/V behavior of a QD is a function of whether the QD is behaving coherently or non-coherently [1]. This nonlinear behavior (either coherent or non-coherent) can be due to tunneling, which is an effect created by the wave-like characteristics of electrons. The measured electron tunneling distance associated with QDs is typically on the order of 1 to 10 nm but can be greater and is a function of the size of the QD, the material that contains the QD and other variables. 

Ferritin is a spherical iron storage protein that is abundant in living organisms. C-AFM tests performed on ferritin cores deposited using a highly disordered layer-by-layer deposition technique have demonstrated similar nonlinear I/V behavior as that seen in disordered arrays of QDs, with a measured current of 0.4 nA at 3 V over distances as great as 40 nm. In addition, anomalous current measurements of 0.4 µA have also been reported for disordered multilayer planar arrays of ferritin that have been deposited between parallel electrodes over distances as great as 40 microns, for an applied voltage differential between electrodes of up to 3 V [2,3,4,5]. This current-voltage behavior of ferritin is unlike the sigmoid-shaped current-voltage response that is typically observed for individual electronic junctions formed from organic material. A comparison of the c-AFM I/V response of a semiconductor QD and ferritin is shown in Appendix A and shows that these I/V curves are similar. Notably, the c-AFM I/V response for both semiconductor QDs and ferritin exhibit: (1) relatively flat current response at voltages up to 0.5 volts per QD or ferritin core (unlike sigmoid curves), and (2) non-linear increases in current levels above a threshold voltage. In view of this c-AFM I/V behavior for individual ferritin cores, it would not be expected that a disordered multilayer planar array of ferritin would conduct electrons at an applied voltage differential of 3 V, much less at a level that is three orders of magnitude greater than the current measured for individual cores over much shorter distances due to tunneling. 

Electron transport through two-dimensional disordered QD arrays has been studied using computer models [6]. These computer models show that the electrical behavior of disordered QD arrays is different from the electrical behavior of individual QDs, and that electron transport occurs at much lower voltage levels due to modeled incoherent tunneling between QDs. The modeled I/V characteristic for a number of different two-dimensional disordered arrays is log-linear above a relatively low threshold voltage, as shown in Appendix A. The model reported in [6] was based on QD arrays between parallel metal electrodes, where the QD arrays have a controllable degree of disorder that is modeled as variable inter-dot capacitances. The model was also based on a temperature of zero degrees, to allow the role of thermal fluctuations on tunneling rates to be ignored. Tunneling from metal electrodes to the QDs necessarily means that electrons are behaving as coherent or non-coherent waves when they tunnel, and they must also be exhibiting wavelike behavior in order to tunnel through the arrays. It appears that the tunneling behavior of electrons through disordered arrays of ferritin is thus similar to the tunneling behavior of electrons through disordered arrays of QDs. As shown in Appendix A, the current measured between two parallel metal electrodes having an interdigitated design with DMFS formed using layer-by-layer deposition was reported to have a log-linear/non-linear response, similar to this modeled QD behavior.

Ferritin comprises a spherical protein shell that contains a complex internal structure that includes both ferrihydrite and ferrihydrite precursors [7], as shown in Appendix A. The protein shell includes light chain ferritin subunits that are able to conduct electrons into and out of the core [8]. Once an electron is inside of the core, the internal ferrihydrite structure is able to trap electrons for microsecond time scales [9]. In addition, the external protein structure of ferritin and the internal structure of the ferrihydrite and ferrihydrite precursors are both chiral, which can contribute to spin selectivity [7,10]. Chirality in organic structures, such as the ferritin protein shell subunits, has also been observed to function as a spin filter, which would increase the spin coherence of exciton electrons generated by individual ferritin cores in disordered layers of ferritin cores [11]. Likewise, the chirality of inorganic structures such as the ferrihydrite core of ferritin has also been observed to facilitate interaction with chiral organic structures, such as proteins [12]. Furthermore, it has been observed that mixed iron oxide phases are present in the ferritin core that include magnetite regions, which could also contribute to spin filtering [13]. These combined effects could help to explain the observed room-temperature electron transport of electrons through DMFS that has previously been observed [14], which could create coherent or non-coherent tunneling at room temperature due to spin coherence that is similar to that modeled in the QD arrays at an assumed temperature of zero degrees [15]. This electron behavior is also consistent with the function of ferritin, which is to store and release iron as part of complex biological processes that utilize iron because it would facilitate iron storage and release under suitable conditions (such as in response to the pH of the environment surrounding the ferritin core) [16].

The electron trapping function performed by ferrihydrite in the ferritin core involves the reduction of iron (III) to iron (II), which may be caused by defects in the crystalline structure of the ferrihydrite, or which may itself cause such defects to form. Because ferrihydrite precursors will be formed by the trapping of an electron by an iron (III) atom in the ferrihydrite crystalline structure and will be located in the immediate vicinity of the iron (II) atom that is subsequently formed, those compounds will be able to reform into the ferrihydrite crystalline structure if the iron (II) atom releases the trapped electron from the 3d^6^ orbital. This process not only results in the release of an electron that has the same spin as other trapped electrons that may be released at or near the same time, but also causes the electron to lose energy with each trap and release event, which increases the de Broglie wavelength of the electron. The coherence length of the exciton (which contributes to the tunneling distance) is directly related to the de Broglie wavelength of the electron. This process could explain why ferritin is able to exhibit the same behavior as QD models at zero degrees, namely, because the role of thermal fluctuations on tunneling is not sufficient to overcome the effect of coherent spin and increased de Broglie wavelength of the electrons. 

Ferritin may thus be able to provide a relatively inexpensive material with a low environmental impact for use in semiconductor devices. The layer-by-layer deposition technique discussed in [14] (and described in further detail below) would be useful for the fabrication of semiconducting devices that use nanoparticles, which are otherwise very difficult to manipulate, and could possibly be used to reduce manufacturing costs for devices that use ferritin or even semiconductor QDs if the process could be modified for use with other nanoparticles. Given the difficulty of both forming devices that use nanoparticles and of verifying the proper construction of such devices, a simple layer-by-layer process that creates repeatable and consistent results would be of potential value. 

The results provided in [14] included an analysis of the work function of the ferritin layers using scanning Kelvin probe microscopy, optical measurements for determination of direct and indirect bandgaps, and other parameters that were consistent with prior results, so those tests were not repeated. However, the results reported in [14] do not provide much detail as to parametric measurements, including the I/V behavior of disordered ferritin layers for one through five layers, the I/V behavior for gaps of different sizes between electrodes, and other important experimental details. Thus, the first objective of these tests was to determine whether the layer-by-layer process could provide consistent results that would allow for the manufacture of commercial devices using that process. 

It is also noted that strongly correlated electron behavior has been reported in transition metal oxides [17], QD arrays [18,19], and twisted double bilayer graphene [17]. Such strongly-correlated electron behavior is a type of strong particle correlation that results when particles interact with each other to create behavior (such as high-temperature superconductivity and switching between a high resistance Mott insulator and a low resistance metallic conductor) that is different from the behavior of individual particles [17]. One of the observable characteristics of strongly correlated electron behavior in twisted double bilayer graphene is collective electron behavior, such as switching from a conductive state to a non-conductive state. This strongly correlated behavior could explain the conductivity of QD arrays, including disordered arrays, and would also predict that such arrays could be “switched” from blocking to conducting as a function of the design of the circuit containing the QD array, which was the second objective of these tests.

In order to determine whether the electron transport that appears to occur in any DMFS that might be formed was due to either: (1) quantum mechanical effects of coherent or non-coherent electrons or (2) classical electron conduction, a four electrode device was used. As shown below, the design is similar to that used in [14], but without the interdigitated structure. In addition, the interdigitated structure from [14] with four electrodes was also tested for comparison purposes. Ferritin was deposited using the layer-by-layer technique discussed in [14], and the current was measured between the electrodes in 5 configurations: 

Configuration (1) with potential applied to electrode A1 and electrodes B1, B2, and B3 in series and connected to ground; 

Configuration (2) with potential applied to electrode A1, electrode B1 connected to ground, and electrodes B2 and B3 floating (to simulate a high impedance ground connection); 

Configuration (3) with potential applied to electrode A1, electrode B2 connected to ground, and electrodes B1 and B3 floating; 

Configuration (4) with potential applied to electrode A1, electrode B3 connected to ground, and electrodes B1 and B2 floating; and

Configuration (5) with potential applied to electrodes B1, B2, and B3 in parallel and with electrode A1 connected to ground. 

If the ferritin layers behave classically, then a linear I/V behavior as shown in Figure 1b would be expected with one-third of the current in each of the three electrodes B1, B2, and B3 as shown in the circuit of Figure 1a for Configuration 1. However, if quantum mechanical electron transport occurs, then log-linear/non-linear I/V behavior would be expected, similar to what was seen in [6,14]. In addition, it would be expected that when two or more electrodes are connected to the ground in parallel by the same impedance, that current flow would be blocked in either direction because strongly correlated electrons would need to flow through the same path or not at all. This would be a macroscopic example of the high resistance state of strongly correlated electrons that has been observed on a microscopic level, which would result because the electrons stored at each ferritin core/QD form a Mott insulator. Otherwise, strongly correlated electrons should follow the same lowest impedance path to the ground, if multiple paths of different impedance were available because a Mott insulator state is not formed in this configuration. For example, the I/V behavior in Figure 1d would be predicted for the circuit design in Figure 1c for Configuration 4.

## 2. Materials and Methods

### 2.1. Ferritin Deposition on Si Surface

The test dies were formed from n-type (As) silicon(111) wafers (0.0025–0.004 Ω cm, 525 µm, SSP Prime from University Wafer), which were heavily doped, and which were cut into small (10 mm × 10 mm) dies. The test dies were then cleaned using a sequence of ethyl acetate, acetone, and ethanol, for 5 min in each. The solvent-cleaned dies were placed in freshly prepared acid piranha solution (H_2_SO_4_:H_2_O_2_ = 7:3 *v*/*v*) for 10 min at 80 °C, then were washed with deionized water. The dies were then dipped in 2% HF solution for 1 min and washed with deionized water, then placed in freshly prepared base piranha solution (NH_4_OH:H_2_O_2_:H_2_O = 1:1:5 *v*/*v*) for less than 1 min at 70 °C, which was carried out to make the wafer surface hydrophilic via oxide layer formation and also to remove any metallic contamination on the die surfaces. The dies were finally rinsed with deionized water and dried under nitrogen gas.

Using the process discussed in [14], alternating disordered layers of cationized (positive) and native (negative) ferritin were formed. The native holoferritin (HoSF) (Sigma-Aldrich, St. Louis, MO, USA) was first diluted to 200 nM concentration using the required amount of MOPS–NaCl buffer (10 mM MOPS (Sigma-Aldrich, St. Louis, MO, USA) in 0.15 M NaCl (Merck, KGaA, Darmstadt, Germany) solution, pH 6.1), then filtered using a 0.22 µm syringe filter and the filtrate was stored at 4 °C. The cationized HoSF was diluted to 200 nM concentration using the MOPS–NaCl buffer, filtered using a 0.22 µm syringe filter, and the filtrate was stored at 4 °C.

Protein monolayers were formed by first incubating the negatively charged silicon substrate in 200 nM cationized ferritin in MOPS–NaCl buffer for 15 min, after which the protein-coated silicon surface was rinsed with deionized water, and then dried under nitrogen gas. For the next layer, cationized ferritin-coated silicon was incubated in 200 nM holoferritin in MOPS–NaCl buffer for 15 min, and the protein-coated surface was then rinsed with deionized water and dried under nitrogen gas. These steps (coating with cationized ferritin and native ferritin) were repeated up to the desired number of protein layers. Samples deposited with ferritin were stored at 4 °C in a nitrogen environment.

### 2.2. Ferritin Deposition on Patterned SiO_2_/Si Surface

Also using the process discussed in [14], test dies were created having dimensions of 1 cm × 1 cm, with a silicon substrate, a silicon dioxide surface, and gold electrodes in the described configurations (with electrodes A1 and B1-B3 in interdigitated and linear configurations). The following layer-by-layer deposition process was then used to form DMFS on these dies. The dies were cleaned with soap water, followed by washing with deionized water and drying under hot air. Then the dies were then treated with acetone at 50 °C for 10 min, then with isopropanol at 70 °C for 5 min, and washed with ethanol and deionized water. Finally, the dies were dried under nitrogen gas and were then ready for protein modification. The layer-by-layer protein deposition method was then used to form DMFS on the electrode dies. Dies deposited with ferritin were stored at 4 °C in a nitrogen environment.

### 2.3. AFM and c-AFM Measurements

AFM and c-AFM was carried out by a Bruker dimension icon AFM. Topography and current of ferritin were simultaneously collected using the contact mode (PFTUNA tip, Bruker Corporation, Tip radius 25 nm, Frequency 70 kHz, Spring constant 0.4 N/m). The image scan rate was set at 1 Hz per line with a resolution of 256 × 256 pixels, corresponding to a frame rate of 256 sec per frame.

### 2.4. Lateral Current Measurements

Electrodes were probed using a station with four tungsten tip microprobes with 10 µm diameter tips, using a Keithley semiconductor characterization system (Tektronix company, 4200-SCS, Beaverton, OR, USA). The voltage was varied from −3 to +3 V to measure lateral current for the circuit configurations shown in Table 1 below.

## 3. Results

### 3.1. Ferritin Morphology on Si Surface

Figure 2a shows a single layer of ferritin formed on the Si substrate. A continuous layer can be observed on the Si surface. A line profile (Figure 2f) indicated that the size of a ferritin structure in this single layer is about 7 nm height, which is similar to the actual size of ferritin as reported [14]. However, the lateral size of an individual ferritin core shown in Figure 2f is ~50 nm, possibly due to the widening effect of the AFM probe. Figure 2b–e show that DMFS were able to be formed using the layer-by-layer technique described in [14]. The ferritin in these structures appears to be fairly continuous and compact (the height variation is for the maximum and minimum measured across the top layer, and not an absolute height measurement relative to the base). The same technique was used for fabricating each of the test dies. Although it was not possible to perform lateral current tests on those devices, c-AFM area scans were performed to measure the I/V properties of these ferritin layers on Si substrate perpendicular to the plane of the layers (as opposed to the point I/V characteristics measured in [14]). The results were shown in the graph of Figure 3, and individual scan images are shown in Appendix A. These results are consistent with previously measured I/V characteristics and show an increase in current with an increasing number of layers up to five layers.

### 3.2. Ferritin on Si Surface Studied by c-AFM

It was determined from the c-AFM tests that the surface topography of one layer of ferritin is not uniform and that some electrical discontinuities can be found on the surface, as shown in Figure 2a and Appendix A. Nevertheless, areas with electrical continuity can still be observed between these areas, see Figure 3 and Appendix A. This is consistent with the results previously reported in [14]. The DMFS formed by the layer-by-layer deposition technique are shown in Figure 2b–e. The c-AFM current measurements obtained from these layers also show an increase in current at voltages greater than 2.0 V and less than −2.0 V for all the devices, which is consistent with previous tests. For 5 layers of ferritin, there is a decrease of current which might be due to the thickness of ferritin that has reached the peak-current value because of insulation.

While the discontinuities did not appear to form large areas without any ferritin, they were still significant enough to raise concerns regarding whether they would prevent the formation of DMFS that were capable of electron transport. This occurred in approximately 75% of the test dies, as discussed below. Thus, although the layer-by-layer technique was partially effective for the purpose of these tests and was able to provide at least some restricted areas where DMFS were formed, those structures do not appear to be sufficiently continuous for use in manufacturing devices when made using the process discussed in [14]. 

### 3.3. Ferritin on Si Surface Studied by Keithley Measurements

The Keithley was used to measure applied voltages and currents on test dies using the shielded microprobes. Tests on dies without any ferritin layers resulted in currents that were generally less than 100 pA. One of the two die configurations with interdigitated electrodes that was tested is shown below in Figure 4a and is referred to as model 1. This die had a configuration similar to the die tested in [14], except for the modification to create 3 electrodes B1, B2, and B3.

A second test die was also made that did not use interdigitated electrodes. The design is shown in Figure 4b. The electrodes A1, B1, B2, and B3 were formed from gold on a silicon oxide substrate. Three different gaps between electrode A1 and electrodes B1, B2, and B3 were used (20, 40, and 80 microns). The electrodes B1, B2, and B3 were isolated from each other with silicon dioxide, as shown by the arrows. The DMFS was formed on top of these dies, as shown in Figure 4c. 

The test equipment configuration is shown in Figure 4d and included shielded leads for contacts to the test die to reduce the effects of noise caused by environmental EFI, and a separate ground connection to reduce DC offset or other grounding effects. Voltage and current measurements were derived from the leads.

The first set of tests on a model 2 die with no ferritin resulted in low current levels. For the first configuration with electrodes B1, B2, and B3 connected to ground and voltage applied to electrode A, the I/V curve shown in Appendix A was obtained. This current appears to be due to a DC calibration offset because there is no variation in current, and also because there is 450 pA measured at 0 volts. It is noted that performing measurements of currents at this level is difficult and that in this case the discrepancy can be explained. Other DC offset measurements were also recorded, and it was possible to account for those DC offsets in the small number of measurements where they were present.

For a second configuration, electrodes B2 and B3 were left ungrounded (float) to simulate a high impedance path to ground, electrode B1 was grounded, and voltage was applied to electrode A1. These measurements appear to be mostly noise below 30 pA, which indicates that any current that was flowing between the electrodes with no ferritin deposited was below the sensitivity of the test equipment. As previously mentioned, measuring current at the level of picoamperes is difficult, and a low level of current between electrodes of the bare dies is expected and consistent with the results reported in [14].

For a third test configuration, electrodes B1 and B3 were left ungrounded, electrode B2 was grounded, and voltage was applied to electrode A1. These measurements appear to be mostly noise below 100 pA, which again indicates that any current that was flowing between the electrodes with no ferritin deposited was below the sensitivity of the test equipment.

For a fourth test configuration, electrodes B1 and B2 were left ungrounded, electrode B3 was grounded, and voltage was applied to electrode A1. These measurements appear to be mostly noise below 30 pA, which indicates that any current that was flowing between the electrodes with no ferritin deposited was below the sensitivity of the test equipment.

For a fifth configuration, electrode A1 was connected to ground, and electrodes B1, B2 and B3 connected to the applied voltage. These results mirror the results from the first test configuration, and also appear to reflect a DC offset because there is no variation in current and there is −550 pA measured at 0 volts. Thus, disregarding the data from the first and fifth test configurations that indicates a DC offset at the current measuring equipment, it appears that the dies without ferritin conduct current below the 100 pA sensitivity of the test equipment. Tests were performed on 19 additional blank dies with similar results.

Tests were then performed on different dies with differing numbers of layers and different gap spacings, but many of those tests yielded current measurements of low-level noise that was below the sensitivity of the test equipment, or other unreliable results that indicated possible damage. Appendix A summarizes these test results and identifies tests where currents of greater than 1 nanoampere with no apparent high-frequency noise components were measured. As can be seen in Appendix A, 8 of 36 tests produced results that were above what appeared to be the sensitivity of the test equipment, or which were not otherwise the results of possible contamination or damage (dies were damaged during a sonic cleaning process that was performed for some dies, which were reused). Thus, it is concluded that while the layer-by-layer deposition of ferritin as disclosed in [14] can produce some useful results, that it is not a reliable process for producing devices with repeatable electrical characteristics. Several of the eight tests that yielded measurable results are discussed further below. 

Several important observations can be made from these results. First, despite the fact that the layer-by-layer ferritin deposition resulted in substantial amounts of ferritin being deposited, many of the tests resulted in measured currents between electrodes of less than 100 pA and essentially equal to the current measured for bare dies without ferritin. As such, those tests establish that merely depositing ferritin in a multilayer structure using the layer-by-layer technique does not cause a measurable change in the measured current. This makes sense, given that the ferritin protein complexes themselves are only 12 nm in diameter and have almost no current measured for applied voltages of less than 0.5 V.

The tests that resulted in currents greater than 100 pA appear to be due to the formation of DMFS that are behaving in a non-classical manner (i.e., with non-linear I/V response) that is consistent with electron transport through disordered arrays of QDs, similar to what was measured in [14]. However, it was not possible to precisely determine the relationship between variations in gap size or the number of layers of the DMFS, due to the apparent inconsistent structure of the DMFS as formed by the layer-by-layer process. Even though these results generally indicated an increase in current with decreasing gap size and an increasing number of layers, they did not allow for conclusions to be made regarding the mathematical relationship of current as a function of parametric variations on the number of layers and the spacings between electrodes to be made. 

The results of specific tests are discussed in greater detail below. 

### 3.4. (A) 80 µm Gap, 4 Layers (Model 2)

The first test to be discussed was on a type 2 die with 4 layers of ferritin and an 80-micron gap. While the logic/objective of the different gaps and layers was to determine the mathematical relationship associated with increasing layers and increasing gaps, the results of these tests were unable to accomplish that objective. However, the eight tests that were effective at generating currents greater than 100 pA with a smooth nonlinear I/V response were able to accomplish other objectives: (1) establishing that layer-by-layer ferritin deposition is capable of forming DMFS, and (2) that the DMFS exhibit strongly correlated electron transport behavior, as shown by path-dependent electron transport behavior such as diode-like switching. 

For test configuration 1, the I/V curve of Figure 5a was obtained.

These results show a substantial current level of −3 µA at −3 V, and about 500 nA at 3 V, with a log-linear/non-linear increase in current with voltage. This nonlinear I/V behavior is consistent with modeled electron transport through QD arrays, as opposed to a linear I/V response that would be expected for classical conduction. 

For test configuration 2 (electrode A1 to electrode B1, with electrodes B2 and B3 floating), the results are shown in Figure 5b. It is noted that these results are essentially identical to the results from configuration 1. The fact that the I/V response for the single current path through A1 and B1 is the same as the I/V response for the current path through A and B1, B2, and B3 in parallel indicates that there was only a single DMFS between A1 and B1, and no DMFS between A1 and B2 or A1 and B3.

For configurations 2 and 3, relatively low-level noise was measured, which also appears to indicate that there was no DMFS that had formed an electron transport connection between electrodes A1 and either of electrodes B2 and B3, see Appendix A. However, it is noted that the magnitude of the current measured between electrodes A1 and B2 was slightly higher (200 pA) than what appears to be the sensitivity of the equipment (100 pA), which may reflect a small amount of electron transport through a DMFS between those two electrodes that did not extend to electrodes B1 or B3. It was not possible to determine the exact structure of the DMFS, though.

For configuration 5, the measured current was similar to configurations 1 and 2, see Figure 5c. The “mirror image” effect of measuring negative currents at positive applied voltages was due to a reversed current measurement direction relative to the applied voltage using the Keithley probes. In particular, the current was measured flowing out to the ground lead instead of into the die when test configuration 5 was used, in order to minimize the number of changes to the test configuration. 

The voltage conventions used for configurations 1–4 were based on electrodes B1-B3 being grounded and voltage being applied to electrode A1. When electrode A1 was grounded and −3 volts was applied to electrodes B1-B3 in configuration 5, it was recorded as +3 volts, and vice versus (i.e., the I/V curves are flipped about the Y-axis for configuration 5). 

These test results show a log-linear/non-linear I/V response, which is consistent with the models in [6]. The reason for the difference between higher levels of measured negative currents at −3 volts and lower levels of positive currents at +3 volts is unclear, but may be due to differences in electron and hole mobility in the DMFS, non-uniform DMFS between electrodes A1 and B2, DMFS between electrodes B1, B2, and B3 that exclude electrode A1 or other variations.

### 3.5. (B) 20 µm Gap, 3 Layers and Retest (Model 1)

A second test that yielded results greater than the equipment sensitivity was a model 1 die with 3 layers of ferritin and a 20 µm gap. For the initial test, the configuration 1 test yielded −9 nA at −3 V and 4 nA at +3 V, with a log-linear/non-linear I/V response curve for negative currents, but a different non-linear response for positive currents, see Figure 6. 

For this second test, configuration 2 yielded noise below 100 pA, but configuration 3 yielded noise that was slightly higher, which may indicate some weak ferritin layer connections between electrodes B2 and A1 for this die, see Appendix A. The fourth configuration also yielded −9 nA at −3 V and 4 nA at +3 V, with a log-linear/non-linear I/V response behavior on the negative current side of the curve, but the different non-linear response for positive currents that is consistent with the behavior of a Mott insulator. Configuration 5 yielded −9 nA at 3 V and 4 nA at −3 V, with an I/V response curve that is nonlinear. Based on these results, it appears that there was only a single good ferritin layer connection between A1 and B3, but that there may have been a second weaker DMFS connection between A1 and B3, which could have resulted in a more complex current distribution, capacitive charging effects or other electrical behavior that had an impact on the I/V response.

This die was then stored for several weeks in an inert, dry nitrogen atmosphere at 4 degrees celsius, and then retested, At the retest, the I/V behavior was different. For configuration 1, the results were −90 to −130 nA at −3 V and less than 5 nA at 3 V for configuration 1 in the initial test, see Figure 7.

The I/V behavior for configurations 2 and 3 also indicated that a change occurred in the I/V behavior of the A1-B1 connection, with an increase from a maximum noise level of 80 pA to a maximum noise level of 180 pA, see Appendix A. For configuration 4, −18 nA was measured at −3 V and 4 nA at 3 V, with a log-linear/non-linear response for the negative currents but a different non-linear response for the positive currents. For configuration 5, the reverse of the I/V measurements for configuration 4 was measured, as opposed to the reverse of configuration 1. It is also noted that the retest I/V measurements provided clearer evidence of Mott insulator type behavior.

### 3.6. (C) 20 µm Gap, 4 Layers (Model 1)

For tests on a model 1 die with a 20 µm gap and 4 layers of ferritin, the current increased to −30 nanoamperes at 4 layers with a 20 µm gap (relative to the first test of the same die with 3 layers, but not the retest), which is consistent with an expected improvement in current with additional layers, see Figure 8a.

For configuration 2, low-level noise was measured, consistent with other measurements of current below the sensitivity of the test equipment. For configuration 3, current in excess of noise levels was measured, with a highly nonlinear I/V response, see Figure 8b. For configuration 4, the I/V response was similar to configuration 1, see Figure 8c. These results also provide evidence of a Mott insulator state providing an impedance-based switching effect, similar to those obtained for the tests in Section 3.5 (B).

### 3.7. (D) Model 1, 5 Layers 40 µm

The results of I/V tests for configuration 1 of a model 1 die with 5 layers of ferritin and a 40 µm gap were −3 µA at −3 V and 2 µA at 3 V, see Figure 9. 

The results for tests on configuration 2 were low levels of noise, consistent with other measurements below the sensitivity of the test equipment for different configurations. For configuration 4, there was a high-level current transient at −1 V that may be indicative of a DMFS configuration that resulted in at least some electron transport, see Appendix A. Configuration 3 resulted in the same I/V behavior as configuration 1. Configuration 5 resulted in an I/V curve that ranged from 900 nA at −3 V to −700 nA at 3 V, in a log-linear/nonlinear distribution. There was no evidence of a Mott insulator state for this configuration. 

## 4. Discussion

The observed I/V behavior of the different configurations of dies and layers provides evidence that layer-by-layer ferritin deposition is able to form DMFS with sufficient order to support strongly correlated electron transport between parallel electrodes that is capable of functioning as a Mott insulator. The I/V behavior also demonstrates that these currents are not an artifact of the ferritin itself, but rather of the ferritin in the multilayered configurations. In particular, when the degree of disorder is too great, then the result is that the currents measured are consistent with the currents measured when there is no ferritin. This is not surprising, because the layers of ferritin are very thin (approximately 12 nm per layer), plus the individual ferritin cores exhibit near-zero conductivity at voltages lower than 0.5 V. When thousands of ferritin cores are placed in series and a voltage of 3 V or less is applied, conventional current flow would not be expected, in light of the measured I/V response of individual ferritin cores. 

However, the substantially higher levels of current that were measured in roughly 25% of the tested dies demonstrate that the layer-by-layer deposition technique is capable of providing at least small areas with DMFS that form a continuous path between two or more electrodes. The electrical discontinuities seen in the DMFS from the c-AFM data show that the layer-by-layer technique does not create an electrically homogenous structure, and these observed electrical discontinuities in the DMFS appear to be sufficient to prevent conduction between electrodes in most dies. However, the observed I/V response at levels of 1 nanoampere or greater demonstrates that electron transport is occurring at levels that are at least one order of magnitude greater than the I/V response of 100 pA or less, the current level measured for bare dies. 

Furthermore, these tests demonstrate that the electron transport was strongly correlated, consistent with the behavior of electrons in QD arrays. In particular, electron blocking was observed for multiple current paths through DMFS that have the same impedance, which is consistent with a Mott insulator state. This would be expected, because the filling of QDs with electrons that creates strongly correlated electrons will result in a Mott insulator if the QD array forms a bottleneck because the two or more paths combine to form a narrower path, resulting in the filling of available orbitals and creation of a Coulomb blockade. In contrast, when there is a single path available that remains constant or that gets wider, the electrons are able to tunnel out of the array and do not form a Mott insulator at a bottleneck. 

For example, in Section 3.4 (A), the I/V data shown in Figure 5 is generally consistent with electron transport in QD arrays. The same current behavior was observed for current flow between electrode A1 and electrodes B1-B3 in parallel and grounded (configuration 1) as was observed for electrode B1 alone grounded with electrodes B2 and B3 floating (configuration 2). In contrast, only low levels of current were measured for electrode B2 grounded with electrodes B1 and B3 floating (configuration 3) and for electrode B3 grounded with electrodes B 1 and B2 floating (configuration 4). Likewise, in configuration 5, the same I/V behavior was observed for configurations 1 and 2. In addition, more current is measured for negative voltages than for positive voltages, which indicates that electron transport is path-dependent and not a function of classical impedance.

For Section 3.5 (B), the data shown in Figure 6 and Figure 7 is consistent with this analysis. For the original test configuration, there was an apparent DC offset for the configuration data, with a log-linear/nonlinear I/V response for negative applied voltage and a zero current response up to 2 volts, after which the current increased non-linearly. For configuration 2, the current was below the noise/measurement sensitivity level of the current meter. For configuration 3, the noise level was slightly higher, which may indicate some low levels of current conduction through DMFS. For configuration 4, the I/V data was similar to the configuration 1 data without a DC offset, and for configuration 5, the I/V data was also similar to configuration 1, with a reversed direction reflecting the relative voltage measurement convention. The I/V behavior indicates that positive currents were blocked up to 2 V, consistent with a Mott insulator, but negative currents were able to flow in the retest configuration. The current increase at higher voltages would occur if the higher voltage was sufficient to overcome the effect of the Coulomb blockade at the bottleneck and to force the electrons through.

This non-symmetric I/V behavior indicates that electrons traveling from electrode A1 to electrodes B1–B3 see a different path and result in different QD configurations than electrons traveling from electrodes B1–B3 in parallel to electrode A1, such as due to different DMFS. QD structures that connect two or more of electrodes B1–B3, but which do not connect to electrode A1, would present electrons at electrodes B1–B3 with multiple paths that combine at a bottleneck, where the electrons are at the same potential and are exposed to the same electric field vector, as shown below in Figure 10 (positive and negative current conventions are for tests in configurations 1–4, and are reversed for configuration 5).

Electrons traveling from electrodes B1 and B2 on the right to electrode A1 on the left in this figure would see different DMFS with a common connection, but each ferritin core would nonetheless be at the same potential and would be exposed to the same electric field vector, at least at low voltage levels. If it is assumed that there are N parallel ferritin cores in the DMFS adjacent to electrode A1 but N + X cores at electrodes B1 and B2, this first configuration would fill QDs where the two paths from electrodes B1 and B2 meet, effectively forming a bottleneck that results in a Mott insulator because the N + X electrons are unable to pass through the filled QDs at the bottleneck. In contrast, electrons traveling from electrode A1 on the left to electrodes B1 and B2 on the right would see a different DMFS, where each ferritin core is at the same potential and is exposed to the same electric field vector. This second configuration could generate electrons that are able to tunnel from electrode A1 to electrodes B1 and B2 without filling QDs, and creating a Mott insulator because the path broadens from width N to width N + X. 

Furthermore, in the first configuration, the electric field vector at electrode B1 would start to diverge from the electric field vector at electrode B2 as the applied voltage at electrode A1 is increased from 0 to 3 volts, resulting in fewer QDs and a narrower path. Thus, at low voltages, the different structures could result in a Mott insulator and block electron transport from right to left, but when the voltage reaches a high enough level to cause the electric fields to diverge, the current could be able to flow through the dominant B1-A1 DMFS. It is not possible to determine the exact configuration of the ferritin layer structures or to easily model the behavior of such structures even it was known, but subsequent testing with more uniform structures should be able to eliminate this unbalanced I/V response.

For the retest, it is unclear why the current decreased from −90 to −130 nA at −3 V (the reverse of configuration 1) to −14 nA at −3 V (the reverse of configuration 4). It is possible that ferritin layer connections between electrodes B1-B3 cause capacitive charging effects to occur during the tests for configurations 2–4 that had an impact on the tests for configuration 5, but it also cannot be conclusively ruled out that probe movement during test configuration charges or other variables had an effect.

For Section 3.6 (C), the data shown in Figure 8 are also consistent with the existence of multiple ferritin layer connections between electrodes A1 and B2, or possibly ferritin layer connections between B2 and either of B1 or B3. Current blocking occurred for configurations 1, 4, and 5, but no current was measured for configuration 2 and a highly non-linear I/V curve was measured for configuration 3. This indicates that both electrodes B2 and B3 had some DMFS connections to electrode A1, but that the connection between B2 and A1 was weaker, and that the structure between B2 and B3 was configured so as to block electron transport up to 3 volts when B3 was grounded but only up to 1 volt when B2 was grounded.

For Section 3.7 (D), the data shown in Figure 9 are unusual because there is a change from the measurements for configuration 1 to the measurements for configuration 5, and there is also not a substantially greater negative current than positive current. While it is unclear why there was a difference between the results for configurations 1 and 5, these results could be explained by a single DMFS connection between electrodes A1 and B2, such that no Mott insulator switching effect occurred. However, the transient at −1 V for test configuration 4 could have resulted in a structural change in a ferritin layer, capacitive charging, or some other effect that caused the I/V response to change. 

## 5. Conclusions

While the layer-by-layer approach to forming DMFS provides inconsistent results, it is sufficient to provide evidence of quantum mechanical electron transport and Mott insulator blocking in such structures. If such electron transport did not occur, then the I/V measurements should be at the 100-pA level or lower, as shown by many test results where such levels were observed. The fact that at least some tests were able to produce non-linear currents at up to five orders of magnitude greater than the current capacity of bare dies is consistent with electron transport through disordered QD arrays, as is the difference in I/V behavior as a function of the direction of electron travel. However, these tests were unable to control layer configurations sufficiently to perform parametric analysis of the difference in layers and distance, and further testing with more control over the DMFS would be needed, either by improving the layer-by-layer deposition technique to prevent the formation of the electrical discontinuities reported here for the first time or by using micromanipulators or other more expensive equipment and time-intensive procedures. 

## Figures and Tables

**Figure 1 materials-14-04527-f001:**
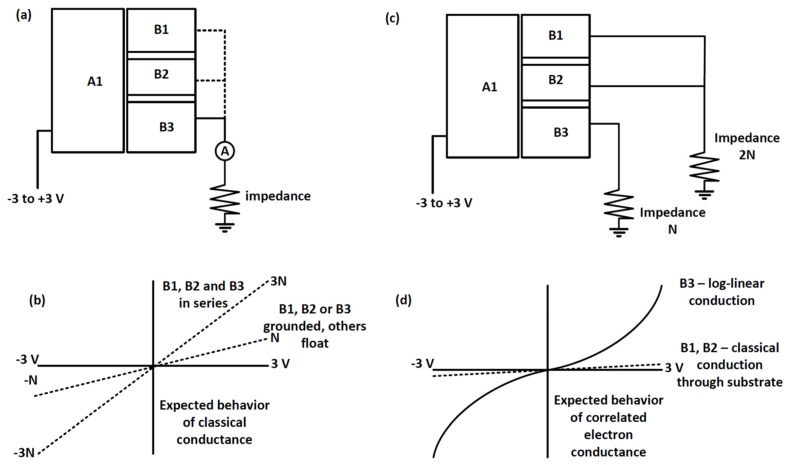
(**a**) Conceptual circuit design showing electrodes connected in parallel to ground through the same impedance (Configuration 1), (**b**) classical I/V current behavior, (**c**) conceptual circuit design showing electrodes connected in parallel to ground through different impedances (Configuration 4), and (**d**) log-linear approximation of coherent electron transport I/V behavior through quantum dot arrays.

**Figure 2 materials-14-04527-f002:**
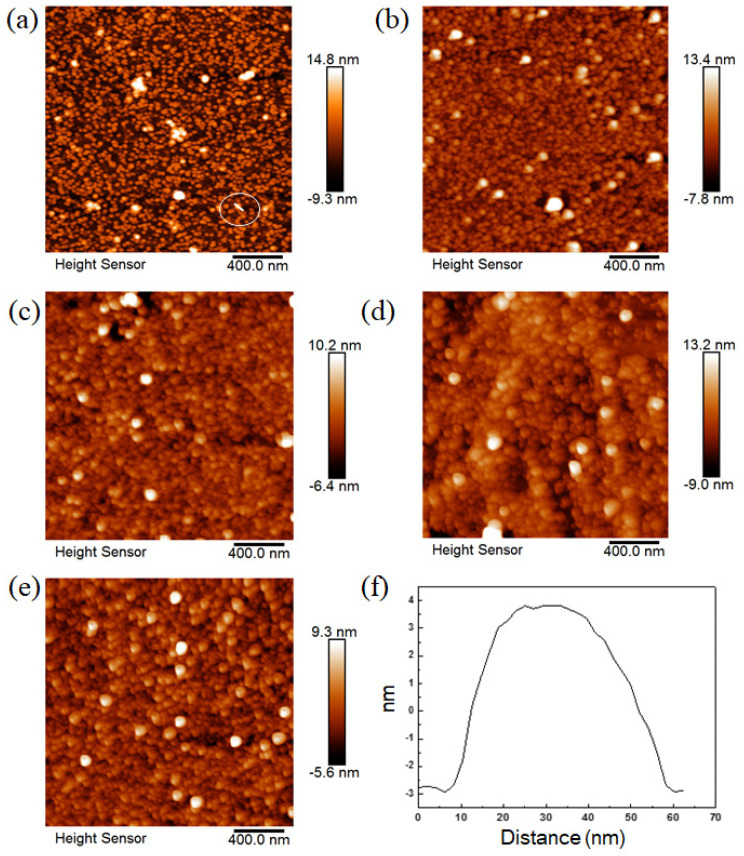
Ferritin layers deposited on a substrate using layer-by-layer deposition. (**a**) One layer; (**b**) two layers; (**c**) three layers; (**d**) four layers; (**e**) five layers; (**f**) line profile of the ferritin as shown in (**a**) indicated in the circle.

**Figure 3 materials-14-04527-f003:**
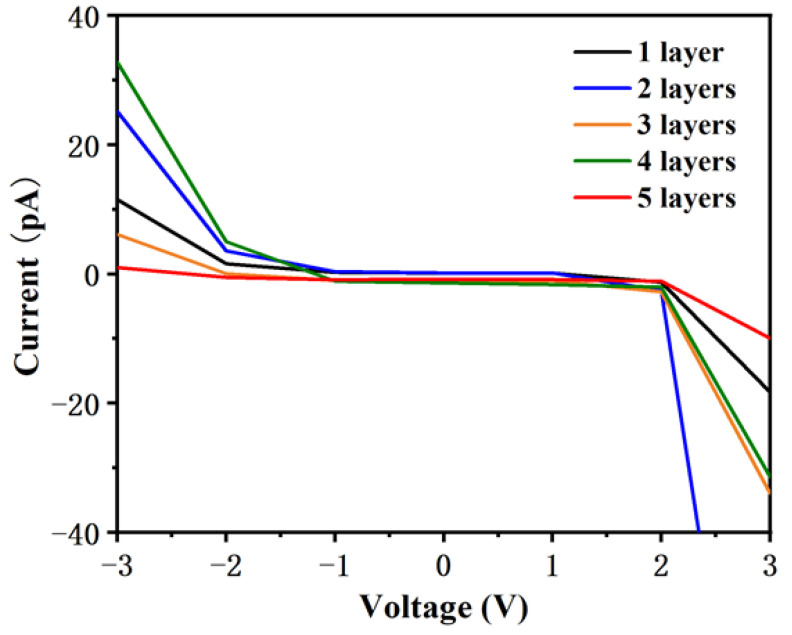
I/V properties of ferritin grow on an Si substrate measured by the c-AFM area scan.

**Figure 4 materials-14-04527-f004:**
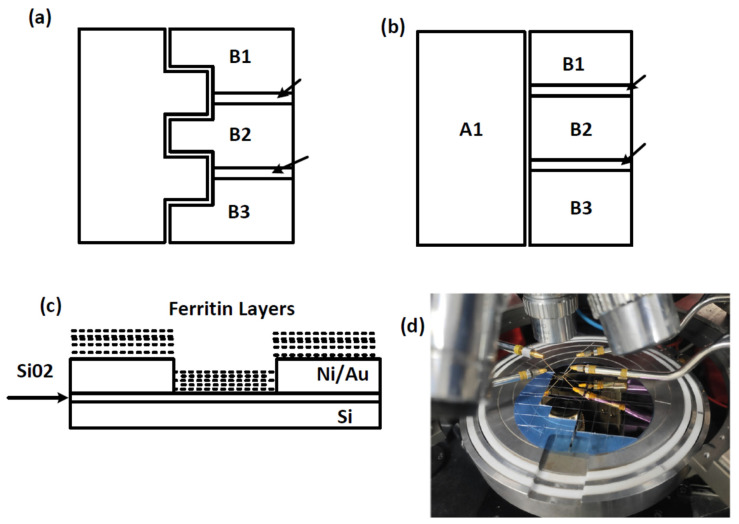
(**a**) Silicon die model 1; (**b**) silicon die model 2, (**c**) side view showing layer-by-layer deposition of ferritin; and (**d**) picture of test configuration.

**Figure 5 materials-14-04527-f005:**
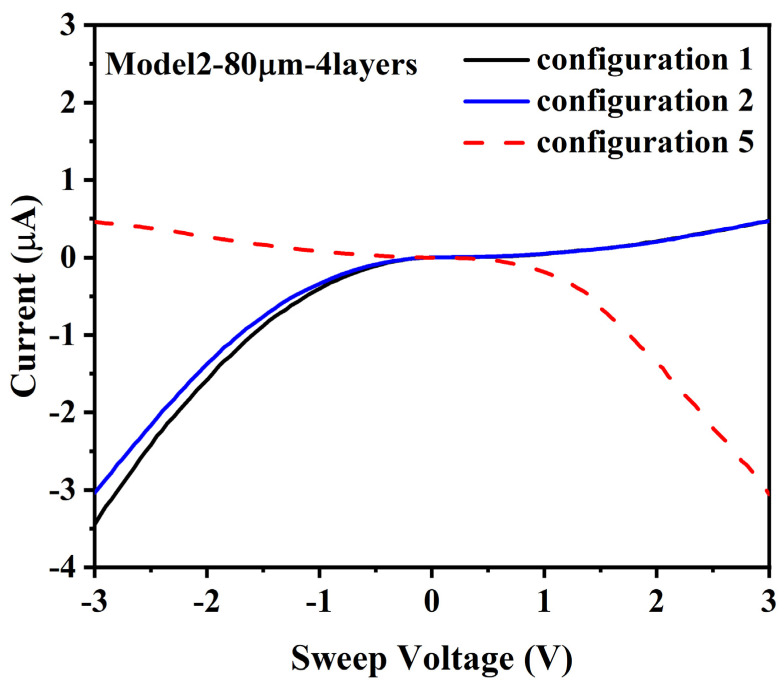
I/V curves for 80 µm gap, 4-layer test specimen: (**a**) (configuration 1) sweep voltage applied to A1, B1–B3 grounded; (**b**) (configuration 2) sweep voltage applied to A1, B1 grounded, B2, and B3 float; (**c**) (configuration 5) sweep voltage applied to B1–B3, A1 grounded.

**Figure 6 materials-14-04527-f006:**
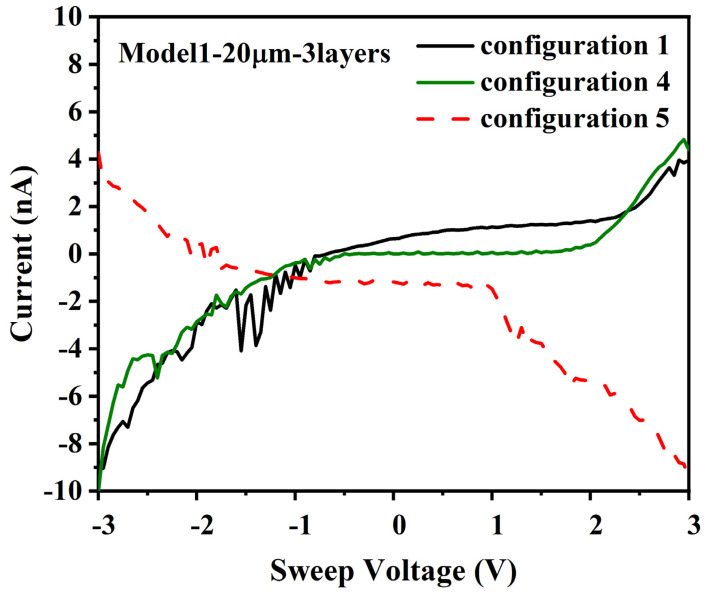
I/V curves for 20 µm gap, 3 layers, and voltage applied to: (configuration 1) A1, B-1B3 grounded; (configuration 4) A1, B3 grounded, B1 and B2 float; (configuration 5) B1-B3, A1 grounded.

**Figure 7 materials-14-04527-f007:**
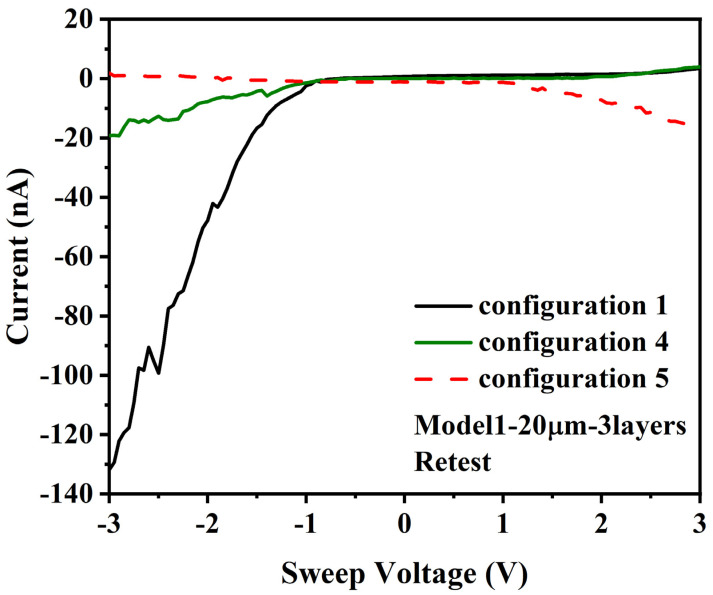
Retest I/V curve for 20 µm gap, 3 layers, and voltage applied to: (configuration 1) A1, B1-B3 grounded; (configuration 4) A1, B3 grounded, B1 and B2 float; (configuration 5) B1-B3, A1 grounded.

**Figure 8 materials-14-04527-f008:**
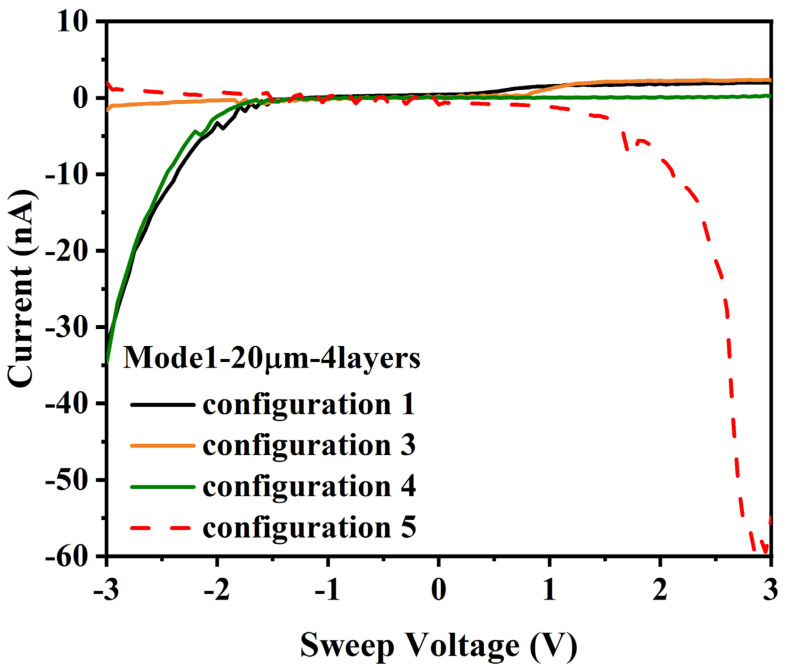
I/V curves for 20 µm gap, 4 layers, and voltage applied to: (**a**) (configuration 1) A1, B1–B3 grounded; (**b**) (configuration 3) A1, B2 grounded, B1, B3 float; (**c**) (configuration 4) A1, B3 grounded, B1, B2 float; (**d**) (configuration 5) B1–B3, A1 grounded.

**Figure 9 materials-14-04527-f009:**
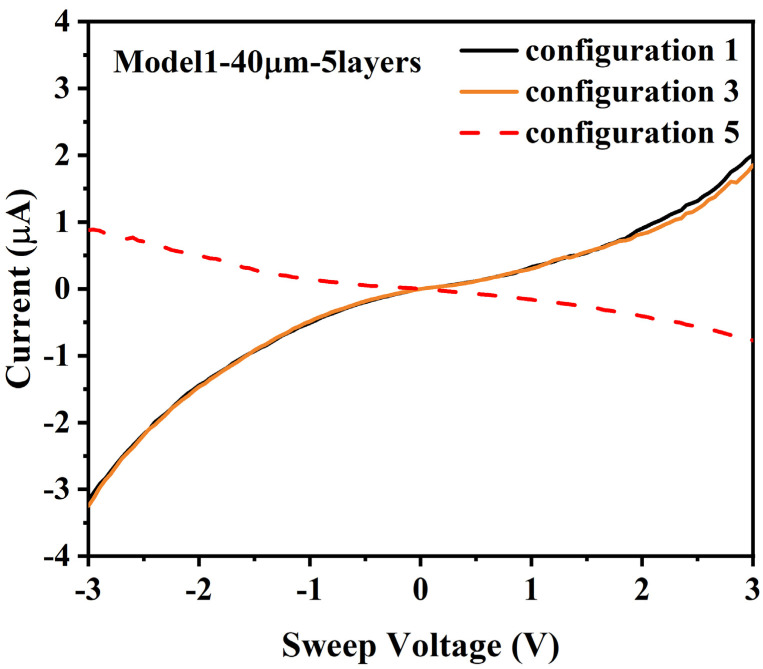
I/V curve for 40 µm gap, 5 layers, and voltage applied to: (configuration 1) A1, B1–B3 grounded; (configuration 3) A1, B2 grounded, B1 and B3 float; and (configuration 5) B1–B3, A1 grounded.

**Figure 10 materials-14-04527-f010:**
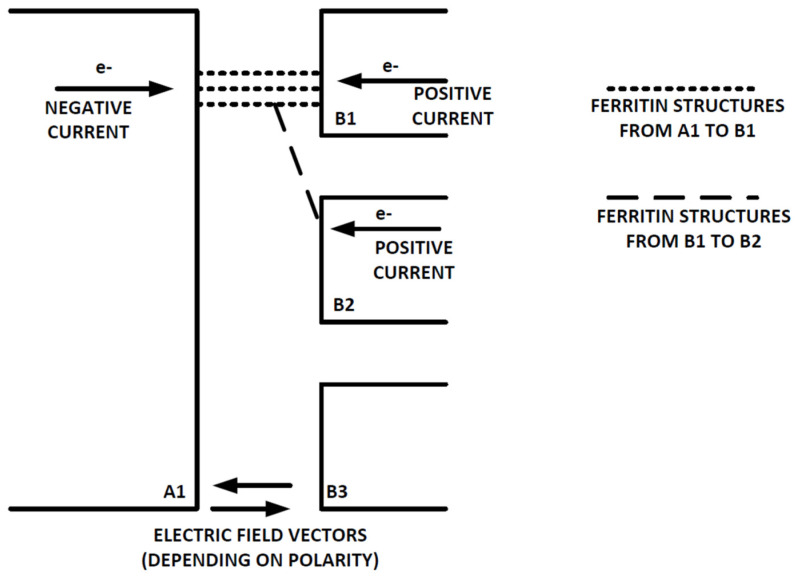
Electron Transport Structures.

**Table 1 materials-14-04527-t001:** Test Configurations.

Configuration	Connection
1	voltage applied to A1, B1-B3 grounded
2	voltage applied to A1, B1 grounded, B2, B3 float
3	voltage applied to A1, B2 grounded, B1, B3 float
4	voltage applied to A1, B3 grounded, B1, B2 float
5	voltage applied to B1-B3, A1 grounded

## Data Availability

Upon request.

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
