# Peer review of "Indication of Strongly Correlated Electron Transport and Mott Insulator in Disordered Multilayer Ferritin Structures (DMFS)"

_materials, 2021, doi:10.3390/ma14164527_

Round 1

Reviewer 1 Report

In this manuscript by Chris Rourk, Yunbo Huang, Minjing Chen and Cai Shen, the electrical characteristics of 1-5 layers of the ferritin arrays are reported. It is expected that various degrees of structural homogeneity can provide a switching function associated with the observed behavior of strongly correlated electrons. But most measurements resulted in noise or damage. The layer by layer approach to forming DMFS provides inconsistent results, some results are interpreted as consistent with the behavior of a Mott insulator and so on. The similarity with QD should be examined by other experimental techniques. There is no control over the structure of the samples. The number of samples is very limited to support the conclusions of the manuscript. More experiments on similar samples are required. The manuscript overlaps with already published papers on ferritin. Highly Correlated Electron Transport should be replaced by Strongly Correlated Electron Transport etc. I cannot recommend this manuscript for publication in Materials.  

Author Response

Comment 1 - “It is expected that various degrees of structural homogeneity can provide a switching function associated with the observed behavior of strongly correlated electrons.”

Response 1 – The various degrees of structural homogeneity do not provide a switching function.  The switching function was provided by the specific configuration of the ferritin cores that results in a Mott insulator forming as a result of a bottleneck for electrons flowing in a direction from a greater number of cores to a smaller number of cores but not the other direction, which results in Coulomb blocking from filling of orbitals.  The research design was appropriate for multiple objectives, where 1) tests on the effect of various degrees of structural homogeneity and 2) switching were 2 of the different objectives.   

Comment 2 – “But most measurements resulted in noise or damage.”

Response 2 – Those results prevented us from determining the effect of various degrees of structural homogeneity on current levels, but did not prevent measurements that clearly show electron transport/switching/Mott insulator.  Those results have not been previously reported and provide a valuable contribution to the associated field of study.  The reported incidents of noise and damage support the conclusion that the methods used to obtain the measurements are adequately described, the results are clearly presented and the conclusions supported by the results, because those results were not excluded as is often done by others and are consistent with the analysis of the other results. 

Comment 3 – “The layer by layer approach to forming DMFS provides inconsistent results, some results are interpreted as consistent with the behavior of a Mott insulator and so on.

Response 3 – This was another observation from these tests that was not previously reported in the literature.  These results will assist others who might be investigating these structures, as nobody has proposed that a Mott insulator could be formed from properly configured arrays. 

Comment 4 – “The similarity with QD should be examined by other experimental techniques.”

Response 4 – The authors agree that further research is desirable, but these results should be published to assist any further work with the lessons learned from this research.

Comment 5 – “There is no control over the structure of the samples.”

Response 5 – A number of parameters were controlled in the structure of the samples – 1) spacing of the gap between electrodes, 2) the number of layers, 3) the electrical configuration of the electrodes, 4) the applied voltages and measures currents, 5) the preparation of the ferritin layers, 6) repeatability of the layer by layer formation process was demonstrated.  To the extent that what the reviewer means is that the layer-by-layer deposition technique resulted in variable configurations, the authors note that the prior literature did not report that and instead appeared to use the single best representative result from an unknown number of samples.  As such, the current results will help to establish expectations about the results of the use of the layer-by-layer technique and the problems that may be encountered if it is used, which may lead to a solution to those problems and the eventual use of the technique for manufacturing in the future once those problems are resolved.  These results will help others to make decisions about future research.

Comment 6 – “The number of samples is very limited to support the conclusions of the manuscript.”

Response 6 – The authors note that the conclusions are carefully qualified as being limited as a resulted of the number of samples.  Those conclusions provide a reasonable inference.  One objective of these tests was to see if the layer by layer process could be used for manufacturing and the conclusion is that it cannot, more tests would not be needed to establish this.  In addition, as another objective was to determine whether the current/voltage behavior for multilayers of ferritin was greater than the current/voltage for a bare die with no ferritin, and this was also accomplished.  Another objective was to determine whether switching would occur under certain circuit configurations, and this objective was also accomplished.  More tests would only be duplicative for most of the objectives.

Comment 7 – “More experiments on similar samples are required.”

Response 7 – The authors agree, and hope that the current paper provides some help to anyone else who might want to perform additional experiments.  However, the experiments should include a focus on improving the layer-by-layer process to increase yield.

Comment 8 – “ The manuscript overlaps with already published papers on ferritin.”

Response 8 – As noted, there are a number of important observations that are made from these tests that were not previously reported and which answer questions raised by the earlier published research.  Any overlap was necessary to provide context for the work done in this paper.

Comment 9 – “Highly Correlated Electron Transport should be replaced by Strongly Correlated Electron Transport etc.”

Response 9 – This has been done.

Reviewer 2 Report

Reviewer # 1:

Rourk et al. investigated electron transport based Ferritin structures with different layers. These structures can provide switching functions. I recommend manuscript to accept after author should clarify these issues.

  • Why I-V characteristics in Fig. 3 shows in the reverse direction. Normally, forward direction in positive side and reverse direction in the negative side.
  • Authors should provide the energy gap of the layers using optical measurements and correlated to the results.
  • How different work functions of metals affect the Ferritin structures.

Author Response

Comment 1 – “Rourk et al. investigated electron transport based Ferritin structures with different layers. These structures can provide switching functions. I recommend manuscript to accept after author should clarify these issues.”

Response 1 – The authors appreciate the approval by the reviewer.

Comment 2 – “Why I-V characteristics in Fig. 3 shows in the reverse direction. Normally, forward direction in positive side and reverse direction in the negative side.”

Response 2 – This was due to the way the test equipment was set up and the changes made to the different electrode configurations.  In order to minimize the number of connection changes that had to be made using microprobes, the current direction for the configuration where the single electrode on the left was grounded and potential was applied to the 3 electrodes on the right in parallel was opposite to the current direction for the configuration where the 3 electrodes on the right in parallel were grounded or tested separately and potential was applied to the single electrode on the left.  This procedure has been clarified in the paper.

Comment 3 – “Authors should provide the energy gap of the layers using optical measurements and correlated to the results.”

Response 3 – This was previously done by others and was not important to the objective of the research.  A note referring to that previous work has been added to the paper.

Comment 4 – “How different work functions of metals affect the Ferritin structures.”

Response 4 – This was also previously done by others and was not important to the objective of the research.  A note referring to that previous work has also been added to the paper.

Round 2

Reviewer 1 Report

The manuscript was revised, it can now be published in Materials. 

Reviewer 2 Report

Accept in present form